# Polyisobutylene and Silicone in Warm Edge Glazing Systems—Evaluation of Long-Term Performance

**DOI:** 10.3390/ma14133594

**Published:** 2021-06-27

**Authors:** Maciej Cwyl, Rafał Michalczyk, Stanisław Wierzbicki

**Affiliations:** 1Department of Concrete and Metal Structures, Faculty of Civil Engineering, Warsaw University of Technology, Al. Armii Ludowej 16, PL 00-637 Warsaw, Poland; mc@il.pw.edu.pl (M.C.); s.wierzbicki@il.pw.edu.pl (S.W.); 2Department of Theoretical Mechanics, Pavement Modeling and Railroad Engineering, Faculty of Civil Engineering, Warsaw University of Technology, Al. Armii Ludowej 16, PL 00-637 Warsaw, Poland

**Keywords:** polyisobutylene, PIB, silicone, glass, dual-seal, warm edge glazing systems, IGU, glass façade, sealing displacement, FE modelling

## Abstract

This article describes the characteristics of one type of sealing system used in warm edge glazing units and analyses the possible causes of damage. Attention was focused on the performance of the dual seal, PIB/silicone system. This type of glazing is widely used for modern curtain walls and roofs of office buildings and shopping centres. Study was focused on PIB displacement defects, which affects both the appearance and thermal performance of the curtain wall system. Wide-ranging field surveys were conducted to examine the problems identified in some office buildings. The information gathered in this way was used to identify the critical areas and causes of seal displacement in the analysed insulating glass units (IGUs). Laboratory tests were conducted on PIB and silicone seals retrieved from the removed defective units. The properties of these materials were determined and used to evaluate the applied edge sealing system and build a representative numerical model. Due to the problems encountered in deriving accurate analytical formulas, finite element (FE) approximation was used as a problem solving tool. The generated FE model and strain analysis were the key parts to obtaining a true representation of the actual behaviour of IGUs subjected to various environmental loads, taking into account the influence of the air cavity. Results of computer simulations and laboratory tests were compared for model validation. The effect of changes in ambient pressure was examined, showing the development of tensile strains in the silicone and PIB, which can lead to debonding. The greatest principal strains occur at the silicone/butyl rubber interface and this location should be considered to be the most susceptible to failure. The observations are summarised in the final conclusions. Additionally, as field study showed, after ten years in service, the percentage of damaged units is considerable. More frequent IGUs inspection should cover both appearance and thermal imaging to detect unsealed panels. From the standpoint of both durability and appearance, dual silicone/PIB should be phased out in favour of modern seal systems.

## 1. Introduction

There is more and more room for the use of glass in the present day architecture. Owing to its properties, glass has become a vital part of large panel shopfronts, shopping malls, high-profile showrooms and, last but not least, residential buildings. Minimizing the consumption of energy has become a worldwide issue and an important goal in the design of buildings [1,2]. With the ever-increasing awareness of the energy efficiency issue, insulating glass consisting of two or three panes combined together to form a single unit of the building envelope has become the material of choice in construction applications [3].

Effective edge sealing is key to obtaining high quality insulating glass units [4]. The traditionally used aluminium spacers create a considerable thermal bridge in the edge seal, compromising the thermal performance of the unit and thus affecting the overall energy performance of the building. Therefore, in order to reduce the heat loss and improve the overall thermal performance of the unit, warm edge spacers, in which metal had been replaced by plastics, were phased in.

Nowadays, glass manufacturers have at their choice a number of different sealing options [5]. Polymers with adhesive properties are the most widely used IGU sealants. The key role of any glass edge sealing system is to provide a structural connection for two or more glass sheets and effectively keep off water vapour, air, dust and other volatile substances from entering the internal cavity (or cavities) between the panes. Furthermore, the sealing must ensure long-term resistance to changing weather conditions, including sunlight, UV and high temperature gradients.

These two main requirements can be satisfied most effectively by a dual-seal configuration [6]. The first component of the system, called the primary seal, is most often based on polyisobutylene (PIB) and serves as a barrier to keep off moisture. The reason why moisture must be kept off is its tendency to condense on the glass surface, which disturbs vision and affects appearance. A very low moisture vapour transmission rate (MVTR) is, therefore, one of the critical properties of the primary seal. Owing to this property, the primary seal will effectively stop noble fill gases (e.g., argon, krypton) from escaping out of the cavity. However, such sealing materials are most often characterised by a very low mechanical strength. Therefore, a secondary seal is added to the system to ensure structural integrity adequate to the expected mechanical loads. Silicone sealants are widely used in this application due to their excellent adhesion to glass and metal surfaces, as well as high elasticity, strength and durability after curing. They are very resistant to extreme temperatures, weather factors and UV radiation. With only one weak point, namely a high permeability to water vapour, they nevertheless ensure high resistance to environmental factors and durability of sealed insulating glass units [6].

While the idea of edge sealing of glass units dates back to the 1940s, it took much time before such glass became commercially available. This technology gained momentum in the 1970s, when glass manufacturers focussed on improving the design of the edge sealing system and the quality assurance methods. Wolf [7] studied the primary environmental factors relevant to the process of ageing of insulating glass. He found that the service life of a sealed IGU critically depends on the perfect functioning of the edge seal exposed to environmental factors (e.g., temperature-induced stress and destruction of the sealants’ adhesion). He also pointed out that gas permeability of the dual-seal is almost solely determined by the permeability of the primary seal because of the much higher permeability of the secondary seal [8].

There is a wide variety of different tests for determining the durability and the expected lifetime of double-pane insulating glass. The history, scientific basis and application of internationally recognised durability tests for insulating glass are presented in [9].

### 1.1. Characteristics of PIB Seal Displacement, ISSUES and Limitations of IGU Durability

While the article considers the overall serviceability of the dual-seal system, particular attention is paid to the failures of the PIB primary seal. Polyisobutylene, a.k.a. butyl rubber, is widely used as a primary seal in the insulating glass edge sealing system. This material owes its popularity to easy machining, high yield and cost-effectiveness. There are many manufacturers of PIB, same as there are many manufactures of windows who use PIB as edge sealant in their end products.

When it comes to composition, the primary sealant formulation typically includes polyisobutylene (PIB), reinforcing fillers, special additives and pigments. Polyisobutylene is chosen due to its excellent moisture and gas diffusion properties. The other ingredients are included in the formulation to enhance the performance properties of the sealant, such as UV and high temperature resistance, rheology and flow characteristics, sheet strength and adhesion to the surfaces of IGU components (commercial grades of PIBs presented in [10]). These ingredients in combination form a gas- and moisture-tight material working in the moving joint between the glass and the spacer edge.

In some windows PIB can be seen in the vision area of the unit which is a matter of concern. Recently, the number of reported structural defects of insulated glass windows has been on the increase. We see an increasing number of claims made by building owners and developers based on visible displacement of PIB sealant from its original position on the edge of the unit into the cavity, where it disturbs the vision area. Tests are therefore necessary to determine whether the seal displacements are caused by material flaws or assembly faults. There are certain environmental conditions (e.g., high temperature or direct sunlight) which may preclude use of this type of insulating glass.

In [11] Watson et al. presented the results of both laboratory and field tests carried out to document the displacements of insulating glass seals. The research was conducted in the United States between 2008 and 2012. The authors focused on determining the frequency of displacement of thermoplastic sealing, including in particular dual-seals including PIB. The analysed displacements resulted in various defects, including twisting, bulging, cracking, out-of-straightness and out-of-squareness of the edge seal. In most cases, the sealing was shifted into the cavity. Alternating inward and outward movements of the seal, indicated by streaks and PIB residue on the glass surfaces, were infrequent. The number of streaks was found to correspond to the number of years in service, suggesting a seasonal nature of these movements. In any case, the result of such extreme seal movements significantly compromised the thermal performance of the glass edge and affected the appearance of glazing. Figure 1 shows examples of seal displacements found in the literature [11]. In addition, the authors of [11] studied cold flow displacement behaviour of various seal spacer assemblies in laboratory tests at various temperatures and under different loading conditions. A four-week long accelerated test showed that PIB-based primary seals are prone to displacement under loading. The reported cause was a poor bond between PIB and the secondary seal.

The National Glass Association (NGA), together with the Insulating Glass Manufacturer’s Alliance (IGMA), recently published the guidelines for the use of PIB primary sealant [12]. The guidelines deal with the problems related to using primary seals in IGU, and particular attention is paid to the performance of polyisobutylene (PIB). Thermoplastic seals used in IGU may be subjected to movements caused by physical and chemical changes.
PIB squeeze out, a.k.a. PIB creep (see Figure 2a)–movement of the PIB past the spacer which results from pressure applied to the IGU edges after or during installation [12].PIB Migration (see Figure 2b)–progressive or continuous flow of PIB into the vision area of the glass that results from a change in rheology (decrease in viscosity) of the material after installation.PIB pumping action–caused by pressure variation in the cavity. The pressure variation is caused by changes in environmental conditions (temperature, barometric pressure, wind load). For example, the fill gas may become very hot in summer, increasing the pressure inside the cavity, a process aggravated by absorbing coating or tinted glass. The edge seals widen up (the edges of glass come apart) and the panes deflect come apart to accommodate the outward acting pressure. When the temperature drops the process is reversed. Such cycling produces the “pumping” effect, in which the seals become alternately stretched and compressed with the changes of weather conditions.

The first two of the above-mentioned problems are caused directly by the material flaws, and therefore will not be analysed directly in this article. Instead, the authors investigated the movements which were most likely caused by the pumping effect.

The above-described “pumping” effect should be analysed as strains caused by mechanical effects. A dual-seal is composed of two parts that are subjected to the same amount of displacement, yet the induced stresses differ greatly due to very different values of stiffness of the two materials. This situation was comprehensively described in [13]. It is noteworthy that this observation led to the development of a new type of glass spacer [14]. As a general design rule, extreme strains induced by the secondary to the primary seals due to the mechanical interaction should be possibly minimized to avoid butyl corruption [15]. Due to incompressibility of both PIB and silicone, any compressive deformation imposed (e.g., by glass panes) to silicone results in lateral stress in butyl.

The above-described problems appear to be aesthetic in nature, yet it needs to be verified if they can compromise the thermal performance of the window.

PROBLEM-1. Over time, the primary seal material (PIB) is subjected to degradation, resulting in cracks which open the path for diffusion of fill gasses. With the diffusion rate of argon being three times higher than that of the air migrating in the opposite direction, the amount of gas filling the cavity will be generating suction applied to the glass panes. This is only one of the possible explanations and the problem needs to be thoroughly investigated.

PROBLEM-2. After installation IGUs are exposed to a variety of environmental factors, such as wind loads, working loads, temperature and atmospheric pressure fluctuations, solar radiation, water vapour. Thermal loads and pressure differences (for example between the manufacturing site and the final location) cause expansion or contraction of the respective IGU components, including fill-gas. In theory, the edge seal, owing to its elasticity, should partly accommodate the resulting stress. The stress is smaller the more compliant the edge sealing is, which by expansion or contraction (as the case may be) accommodates part of the load resulting from the pressure difference. This, however, happens at the expense of the seal strength and excessive strains may lead to irreversible displacements.

### 1.2. Field Study

This research was triggered by the authors’ awareness of problems in long-term use of the above-described IGU sealing systems. An example of such problems were considerable and frequent deformations of insulating glass seals on many windows a large office buildings in Warsaw after about 10 years of service. A visual examination was carried out and sealing defects were found in more than 30% of the total number of 4500 evaluated units.

One of the removed (replaced) windows is shown in Figure 3. This example shows a considerable displacement of PIB seal, which in this case was sucked into the cavity. The characteristic wavy deformation of the inner seal, found in many units, extended over 5–60 cm in length. The wave extended to 2–15 cm into the cavity. The observations showed that the depth of inward deflection of the butyl component changed over the years, as manifested by the traces left by the seal on the interior surfaces of the glass panes (see pumping action [12]).

This problem occurred in many glazing units, irrespective of their shape or dimensions. It was estimated that 10 years after installation, problems of this kind occur in up to 30% curtain wall glazing units. Investigation of the causes and attempts to predict the development of such defects were necessitated by large number of glazing units with similar defects, the probably progressive nature of such defects and a high cost of replacement of these components.

Besides adhesion between spacers and glass, gas tightness is also an issue of primary importance. A complete sealing failure was not found in a majority of the examined units (over 95%). Despite the inner seal deformation, some amount of noble gas (argon) remained inside the cavity. Infrared thermal imaging was employed to determine the thermal performance of the tested glazing units. A FLIR Systems AB infrared camera, model T1020 was used, with a pixel count of 1024 × 768 px and a sensitivity of 0.02 °C to 30 °C. The tests were carried out as recommended in ISO 6781:1983 “Thermal insulation–Qualitative detection of thermal irregularity in building components”. The test was done in mid-November at an outdoor temperature of −1 °C. In the tested building, the temperatures during the measurements were stable and slightly varied (depending on the location) from 22 to 24 °C. The difference between the external and internal temperatures was in excess of the ISO requirement.

In most glazing areas, no significant temperature differences were observed, but on some portions a significant number of glass panes were found to deviate from the others in terms of surface temperature. In some units, a lower temperature was obtained in the middle, which indicates a concave deformation of glass panes, and most probably also sealing failure and diffusion of air in place of argon.

The identified defects were compared with the thermal imaging results to reveal a strong correlation between low temperatures on the surfaces of the glazing units and the presence of defects (including in particular pulling of seals into the cavity by negative pressure). Pulling of warm edge seals into the cavity can compromise the thermal performance of the glazing unit. We can hypothesise that this process will continue and aggravate the problem.

Furthermore, it was established that this problem was much more frequent on south and south-east facing walls due to more sunlight. There, the number of damaged IGUs was up to 60% higher than on walls receiving less sunlight. Therefore, while having some potential effect, the surface temperatures are not considered the primary cause of the damage. High temperatures which develop inside IGU due to sun exposure induce thermal stresses, aggravating the situation and affecting the sealing system durability and thus the overall reliability of the unit. Furthermore, moisture vapour transmission rate (MVTR) of PIB increases with increasing temperature. MVTR of 0.1 g/(m^2^d) at room temperature increases to almost 10 g/(m^2^d) at 80 °C [3]. For this reason, it was decided to experientially determine the performance of silicone-butyl seals at different temperatures (below freezing, room temperature and extreme temperatures) to consider changing exposure conditions.

### 1.3. The Aim of the Study

For the building owner it is of primary importance that IGUs maintain their initial performance for as long as practicable. Failures of glazing not only entail very high replacement costs but also cause much disturbance to the occupants. At the time of launching a new PIB sealing system, the manufacturers are not able to carry out long-term reliability tests. For this reasons, some systems may be more prone to the problems described above than others. This problem may soon become more acute due to the rapidly growing number of office buildings in which IGUs were used as the main type of glazing.

The objective of this research project was to develop a method for predicting the mechanical performance of the dual-seal system under analysis. It is difficult to derive analytical formulas due to the highly non-linear behaviour of the sealing materials and IGU. Several attempts to develop such analytical models have been made and reported in the literature, and thus can be used in assessing IGU durability (e.g., [13]). However, these models are usually based on certain specific assumptions, and thus they are not considered universally applicable. This being so, there is a need for a reliable computational model for the durability assessment and design purposes, with finite element modelling used for stress analysis [16,17]. With this model in hand it would be possible to identify areas where high stress and excessive strains can be expected and implement appropriate engineering measures to avoid failures. However, during literature study authors were not able to find any examples of FEM application in analysing IGU seal displacements. Therefore, an attempt has been made to pinpoint weak points in IGU design and identify the causes of seal displacement, the factors relevant to their functionality, safety of use and aesthetics.

This article is structured as described below. Section 2 presents an example of IGU assembly. Next, experimental studies carried out to determine the basic mechanical properties of the materials are described. In Section 2.3, a finite element model is built, based on the mechanical properties established in Section 2.2. Hyperelastic constitutive modelling was chosen as appropriate to the materials. Next, models of two types of IGU shapes representing different geometries are built.

In Section 3, the results of FEM analysis are described and applied, and the modelling methodology is validated through full-scale bending test of a rectangular panel prototype, which was continued to failure. Section 4 gives a discussion of the obtained results. The results of the research and analyzes served to identify the phenomena occurring in the dual sealed IGUs during long-term use and to determine the probable causes of failure. This section also includes final conclusions that summarize the key research findings.

## 2. Materials and Methods

### 2.1. Design of Tested IGUs

A typical dual seal IGU was used in the test, with the seal composed of PIB and silicone sealants. Glazing included a 6 mm thick ESG-type toughened glass outer pane and laminated inner pane. The latter was composed of two glass panes separated by a PVB interlayer, giving a total thickness of 8.76 mm. The width of the cavity, i.e., the distance between the opposite panes, was 16 mm and the cavity space was filled with argon. The 16 mm high seal spacer was composed of PIB -29 (Kömmerling) seal and DC3362 silicone. The PIB primary seal was more or less 8.0 mm wide over the whole perimeter, while the width of the secondary, silicone seal varied between 10.0 mm and 12.0 mm. Figure 4 shows the pictures of the IGU model and the actual IGU.

Specimens of two shapes were prepared for the tests: an approximately 0.6 m × 1.5 m rectangular unit and a triangular unit with 2.1 m long legs and 3.08 m long hypotenuse). These two shapes are shown in Figure 5.

### 2.2. Properties of Materials

In order to develop and calibrate the constitutive models of the materials under analysis it was indispensable to determine their mechanical properties first. This concerns in particular PIB and silicone due to their non-linear load behaviour and varying deformation depending on the chemical composition and production lot. For this reason, in the first step the materials were subjected to tension and compression at different temperatures. Additionally, the specimens were examined for uniformity of the material. All materials used in the experiments have been extracted during the field study. All the tests were carried out using an Instron 5567 test frame (Instron, Norwood, MA, USA), which is shown in Figure 6a. Taking account of the problems involved in testing of hyper-elastic materials, the specimens were secured in the machine using appropriate Instron pneumatic side action grips, as shown in Figure 6b. The specimen end was clamped by grip jaws controlled by an integrated pneumatic piston. The clamping force is controlled by adjustment of pressure and was maintained at a constant level despite contraction of the specimen during the test owing to built-in compensation feature. This method eliminated the error caused by slippage of specimens from the grips. The rate of displacement was varied for the different specimens between 0.5 mm/min and 70 mm/min.

#### 2.2.1. Testing of Silicone

Carbon black–filled silicone rubber type DC 3362 was used in the tests. Quasi-static uniaxial tension tests were carried out on an INSTRON testing machine operated in displacement control mode. The purpose of the tests was to determine the stress-strain relationship under tension load at different test temperatures. The silicone specimens were 10 mm × 16 mm with 10 cm gauge length (see Figure 7). The test temperatures were: −20 °C, 23 °C and 80 °C. Temperatures higher and lower than the ambient temperature were obtained by placing the specimens in an environmental chamber for approximately 30 min. The test procedure included four loading/unloading cycles (to obtain elongation of ca. 80% of the gauge length), followed by subjecting the specimens to tension until failure. A total number of 28 silicone specimens were tested, five at −20 °C, eight at 23 °C and fifteen at 80 °C. The exact numbers of specimens and test conditions are given on the left in Table 1.

#### 2.2.2. Testing of Bond Strength between Silicone and PIB Seals

The purpose of the test was to determine the force required to break the bond between the silicone and butyl seals and to determine the mechanisms of failure in different temperature ranges. One of the approaches to determining the properties of joints between two different materials is the use of specially designed joint geometries representative of the actual, in-situ conditions. Reliability of results is conditional on obtaining pure stress state with even distribution of over the interface without any stress concentrations. This is achieved by appropriate test geometry. Ideally, the specimens defined in the test protocol should be simple and easy to prepare. The breaking force should, in principle, remain constant with any variation attributed to changes in adhesive strength.

Forty one specimens were cut out from the tested dual seals composed of silicone bonded to butyl layer. The cross-sectional area (16 mm × 18 mm, please see Figure 4) and gauge length (50 mm) were determined for each specimen before testing. The cuttings were then glued with an epoxy adhesive to the T-shape handles. The test set-up and orientation of the actual specimen in the strength testing machine are shown in Figure 8.

The specimens were tested at temperatures of −20 °C, 23 °C and 80 °C. For tests at elevated and reduced temperatures, the specimens were heated/cooled in the chamber for approximately 30 min. The rate of displacement varied between 0.5 mm/min and 35 mm/min. The numbers of specimens and the test conditions are specified in detail in Table 1 on the right-hand side.

### 2.3. Material Characterization Results

#### 2.3.1. Silicone

Figure 9 shows selected results for stretching of the silicone under analysis, represented by engineering stress-strain curves. The data were obtained at various temperatures by multiple loading and unloading cycles for each strain level. The percentages given in the legend indicate how much of the failure load was applied in a given cycle.

Owing to the use of cyclic loading, effects such as hysteresis and the Mullins effect were detected. Some viscoelastic behaviour of silicone was manifested by variable stiffness. These variations are not big enough to affect the sealing performance. Secondly, every time an elastomer is stretched to a strain level higher than it has experienced so far, the response curve changes as a result. The behaviour of a repeatedly stretched elastomer is represented by the curves in Figure 9. The maximum strain level attained by the elastomer because the elastic response of rubber is a function of the strain it has experienced. This is due to the phenomenon of stress softening, known as the Mullins effect [18]. When an elastomeric material (silicone in this case) is subjected to uniaxial tension from its natural state, unloaded and finally reloaded, the stress required on reloading is less than that during the initial loading for elongations up to the maximum elongation achieved during the initial loading.

As it can be seen in Figure 9, the loading/unloading cycles from any given strain level are not represented by a single curve, and some hysteresis is involved. Some degree of permanent set upon removal of the applied load is also observed. However, the significance of these effects is minor and the response appears to stabilise after a certain number of cycles. It is possible to tell apart the overall stiffness characteristics and the Mullins effect. The envelopes of the first loading curves for different strain levels are represented by the dashed lines on the graphs in Figure 9. Other results of the respective tests can be used to evaluate the magnitude of the Mullins effect.

The effect of the load application rate on the behaviour of silicone was examined by stretching the specimens at different rates (see Table 1). Comparing the corresponding stress-strain curves one does not see any considerable difference between different loading rates. Therefore, the behaviour of silicone was assumed to be independent of the strain rate, at least in the range considered for the analysed IGUs.

The ultimate, failure load, caused rupturing of all the tested specimens. Most specimens broke apart at a point located within the gauge length, except for a few cases where failure occurred in the grips region. The maximum strains at failure also varied strongly. Repeatability of results at the same temperature conditions or loading rate was not obtained. However, it should be noted that the specimens were cut out directly from the tested windows, which is not the sample preparation method prescribed by the standard. Therefore, defects and non-uniformity of the material cannot be excluded, which were evident at failure. As a matter of fact, in reality IGU seals are never exposed to such extreme breaking loads, and hence the test results were not directly used for the modelling purposes and are given for reference only.

#### 2.3.2. Polyisobutylene to Silicone Adhesion

Very interesting results were obtained from the testing the adhesion between the silicone and PIB sealants. With very low stiffness polyisobutylene shows a very high sensitivity to both the temperature and the loading rate. As the temperature increases, the adhesion between silicone and PIB decreases, and the material becomes soft and undergoes plastic deformation. Additionally, a strong dependency between the strain and the load application rate was observed. This can be easily seen in Figure 10 showing displacement-force graphs at 23 °C (Figure 10a) and 80 °C (Figure 10b). Depending on the applied rate of displacement, the obtained forces vary by a few tens of percentages. The graphs show only the results obtained for uniform viscoplastic failure of butyl. Part of the forty one specimens had previously been damaged by partial or complete separation due to bond failure between silicone and butyl.

The failure mechanism is temperature dependant. At −20 °C PIB sealant is highly rigid and resistant to load, producing good bond quality. At 23 °C the material becomes more ductile and three types of damage were distinguished. Firstly, viscoplastic flow and cohesive failure of PIB occurred, as shown in Figure 11a. Secondly, adhesive failure occured, as shown in Figure 11c. In some cases, viscoplastic damage of butyl was only partial and accompanied by partial separation due to bond failure, as shown in Figure 11b.

At 80 °C worsening of consistency and failure of the butyl seal were the dominant failure mechanisms, with silicone-PIB seal separation occurring at times. Attention should also be paid to the significant worsening of the shape, as well as to compromised adhesion of PIB sealant in elevated temperatures. The laboratory tests confirmed the accuracy of the information gathered upon review of the literature on the properties of PIB and silicone sealants, as described at the beginning of this article. More importantly, the mechanical properties of the materials were accurately determined and the bond failure mechanisms were examined at different temperatures.

## 3. Development of a Finite Element Model

The previous sections identified the problem of IGU sealing failure based on literature and in-situ testing, namely a survey of defects. In addition, laboratory tests were carried out to determine the behaviour of these sealing materials at different temperatures. In this way, it was possible to put forward hypotheses as to the cause of the problem and make predictions as to the further progress of damage.

Testing the validity of hypotheses with the use of numerical models is way more convenient and far less expensive than with experimental methods. For example, it is possible to study local stress distribution and predict potential stress and strain concentrations. There are no standard calculation methods for IGU details and FEM seems to be the most reliable calculation tool for determining local stress states. For this reason, some outcomes of this research project were also derived from finite element (FE) simulations.

### 3.1. General Input Assumptions of FE Analysis

Finite element models were created and solved using ABAQUS software [19], which is an engineering analysis software package with extended features for analysing mechanical behaviour of glass and rubber materials. While most IGU research projects have focused on durability, thermal and sound insulation performance or the aspects of façade applications, much less attention has been paid to the structural performance of IGUs, including assessment of the bearing capacity of spacer frames during initial loading [20]. Primary and secondary sealing for IGUs is made of materials (elastomers) that can undergo considerable deformations that go beyond the bounds of the classical small strain theory. Consequently, in order to adequately model the displacements and deformations, it is necessary to take into account non-linear kinematics (available under the NLGEOM option of ABAQUS) and non-linearities of materials.

### 3.2. Constitutive Modelling of Materials

#### 3.2.1. Glass and Polyvinyl Butyral (PVB) Interlayer

Elastic behaviour of glass before breakage is well-known and represented by a linear elastic model, with a modulus of elasticity of 70 GPa and a Poisson’s ratio of 0.23. Glass and PVB are considered isotropic and homogeneous materials. The PVB interlayer can be described by various material laws. All of them adequately describe the mechanical behaviour of PVB, depending on the loading patterns. The viscoelastic model is the most general one for time-varying loading. The properties of standard PVB used in experiments are well described in the literature [21], e.g., by the shear relaxation module and the temperature shift coefficient. A constant value of the bulk modulus of elasticity K = 2.0 GPa can also be assumed (the case of highest density polymers). According to [21] the immediate shear modulus is G_0_ = 146.12 MPa and the long-term shear modulus is G_∞_ = 0.15 MPa. Because of the static nature of the tests carried out as part of this research, a constant strain rate can be expected in the interlayer. The behaviour of the material was approximated as linear for a given temperature and strain rate. Therefore, PVB was represented by a linear elastic material, with E = 427 MPa and v = 0.46.

#### 3.2.2. Primary Seal-Polyisobutylene (Butyl Rubber)

The primary seal creates a tight barrier to water vapour and also fill gas (argon, in this case). It is worth mentioning that diffusion, the main component of gas transfer through the primary seal, depends exponentially on temperature, hence the lowest resistance to atmospheric loads in summer [13]. The most important properties of butyl rubber sealant include low density of (*ρ* = 1.09 g/cm^3^ at 20 °C), very low water vapour transmission rate (<0.1 g/m^3^ a day) and high gas permeation rate (<0.002 g/m^2^ an hour) [13].

This material is highly viscoplastic and its mechanical properties, particularly stiffness, vary depending on the temperature. The values of the strength parameters were adopted on the basis of laboratory tests. Figure 12 shows examples of uniaxial tensile (engineering stress-strain) test results as a function of temperature. On this basis an elasto-plastic material model was adopted, including Huber-Mises-Hencky yield condition.

#### 3.2.3. Secondary Seal–Silicone Rubber

The secondary seal ensures proper adhesion, shape and long-term performance of the entire insulating glass unit. The silicone sealants analysed in this research are made of poly-diorgano-siloxane polymer, a material with elastomeric characteristics. In order to create an accurate computational model of the glazing unit it is necessary to create a reliable model to represent the mechanical behaviour of the silicone rubber sealing. Most commonly, hyper-elastic material models are used to describe the stress-strain behaviour of rubber [22,23], as they are highly capable of representing the non-linear response and near incompressibility of rubber.

The main purpose of this part of the research was to propose and identify the most adequate hyper-elastic model for the selected materials. The behaviour of the material can be described in terms of a “strain energy potential”, which defines the amount of strain energy stored in the material per unit of reference volume as a function of the strain. There are several forms of strain energy potentials available in Abaqus to model nearly incompressible isotropic elastomers.

Having a large amount of experimental data, the Ogden and Van der Waals forms are preferred as the most accurate in fitting the experimental results. If only a small amount of test data are available for calibration, the Arruda-Boyce, Van der Waals, Yeoh, or reduced polynomial forms should preferably be used to obtain a reasonable approximation [19]. Only one set of the uniaxial tension data was available to the authors. The Marlow form is usually recommended in such situations (see Figure 13) with a strain energy potential constructed to exactly reproduce the test data, with the behaviour reasonably approximated by other deformation modes.

Hyperelastic constitutive models are often defined in terms of principal stretches, as this class of model has been demonstrated to accurately predict the elastic response of rubber:(1)$$\lambda =(1+\varepsilon_{c})=\left(1+\frac{\Delta L}{L_{0}}\right)$$

In the case of the uniaxial tension-compression test used to determine the model parameters, the principal stretches are expressed as $\lambda_{1}=\lambda ;\;\lambda_{2}=\lambda_{3}=\lambda^{-0.5}$, where $\lambda_{1}$ is a stretch in the direction of load application. The squared principal stretches and associated principal directions are found from the Cauchy-Green deformation tensors, where they are equivalent to eigenvalues and eigenvectors respectively. The strain energy function is additively decomposed into the volumetric and isochoric contributions U(J) and $W(\lambda_{1},\;\lambda_{2},\;\lambda_{3})$. The latter of them is computed using the deviatoric components of the principal stretches defined as:(2)$${\bar{\lambda }}_{i}=J^{-1/3}\lambda_{i},\;J=\det(F)$$
where *J* (total volume ratio) is calculated by the determinant of the deformation gradient tensor.

The Marlow strain energy potential has the form:(3)$$U=W({\bar{I}}_{1})+U(J)$$
where ${\bar{I}}_{1}$ is the invariant of the left Cauchy-Green strain tensor:(4)$${\bar{I}}_{1}={\bar{\lambda }}_{1}{}^{2}+{\bar{\lambda }}_{2}{}^{2}+{\bar{\lambda }}_{3}{}^{2}$$

Figure 13 shows an example of uniaxial stretching of silicone at different temperatures. Using the results of laboratory tests, the optimization procedure for coefficient selection was carried out. Validation was conducted by numerical simulations representing the uniaxial test and the behaviour of the models compared with the test results.

#### 3.2.4. Gas-Filled Cavity

In an IGU the glass panes are separated by a gas-filled cavity. Argon and krypton are preferred over air because of lower thermal conductivity and better dynamic viscosity. Krypton outperforms argon in terms of insulating properties, and higher reduction of heat transfer, yet is more expensive. In the analysed units, argon was used, which is the most popular fill gas. The units were subjected to varying temperatures and changes of gas volume and ambient pressure. These three factors affect the gas fill in different ways [24]. The relationship between them is expressed by the ideal gas equation:(5)pV=nRT
where:

*p*–gas pressure,

*V*–gas volume,

*n*–amount of gas,

*R*–gas constant of R = 8.314 [J/(molK)],

*T*–temperature.

Because the cavity is hermetically sealed, the amount of fill gas remains constant throughout (*nR* = const). In an isothermal process the ideal gas law can be expressed by the equation *pV* = const., i.e., the energy stored in the gas is completely converted to mechanical energy. Since the fill gas is hermetically sealed, a volume change will change the pressure of the gas. The above-mentioned volume change results from the deformation of the loaded panes. Any change from the initial temperature and pressure levels will entail a volume change, i.e., bulging of the glass panes. The initial state of an IGU depends on the fill gas parameters during the IGU assembly process (both place and time are relevant). For the purposes of this research, the following initial fill gas parameters were assumed: pressure of *p* = 101,325 kPa and temperature of *T* = 20 °C.

Table 2 gives the coefficients of the gas model equations as a function of temperature [25]. Thermal conductivity, dynamic viscosity and heat capacity as a function of temperature are given for three fill gases: air, argon and krypton. Molecular weights are also given in the table.

Three phases were distinguished: initial phase (assembly), beginning of the analysis and final phase. For the sake of simplicity, phases 1 and 2 are assumed to be identical (initial gas parameters are the same as after assembly). Any ambient pressure or temperature change during simulation, e.g., due to seasonal or daily variation of temperature, will change the gas condition according to the following ideal gas law equation:(6)$$\Delta p=\frac{nR(\Delta T)}{V_{0}}$$

In view of the above, to accurately predict the mechanical response of IGU the model should treat it as a gas-filled structure. The interaction between the deformation of the structure and the pressure exerted by the contained gas is a primary difficulty that needs to be overcome. ABAQUS includes a feature to represent gas-filled cavities by a “fluid cavity interaction” and surface-based volume definition. The fluid behaviour is based on a pneumatic model (ideal gas), the fluid is compressible, and its volume is a function of pressure and temperature [26]. This numerical model is able to consider the gas cavity effects between the sheets of glass (panes). Additionally, a possible load transfer between the glass panes (i.e., load sharing effect) across the cavity is considered in this model.

### 3.3. Assembly and Solving of FE Model

Finite element simulations were performed in ABAQUS/Standard (version 6.16, Dassault Systèmes, Villacoublay, France), a commercial software package which is appropriate for finding numerical solutions to non-linear problems. A few three-dimensional models of different shapes were used to simulate the IGU behaviour. Examples of rectangular (617 mm × 1517 mm) and triangular (2193 mm × 2148 mm) shapes are shown in Figure 14 and Figure 15. Each model consisted of three main parts: top pane, spacer frame (silicone + seal) and bottom pane. Simple-support boundary conditions were applied to the lateral surfaces at the glazing unit perimeter.

The typical FE model consisted of quadrilateral shell finite elements representing the glass pane (S8R element type from the ABAQUS library [19]). Glass mesh size and pattern were set in the form of an eight-node regular scheme. Laminated glass is characterized by a significantly larger stiffness variation across the thickness of the unit (glass-PVB-glass). In this study, the laminated shell model was used instead of the monolithic model based on the effective thickness approach, which is widely adopted in the current design standards because of its simplicity. The ‘Composite-layup’ available in ABAQUS allows accurately reproducing the through-thickness behaviour. The accuracy of this approach is demonstrated e.g., in [27], where typical two-ply problems are studied. Moreover, the efficiency of this approach is compared with the effective thickness method and 3D continuum models. It was found to provide an accuracy level comparable with 3D continuum models, yet at a much shorter computation time.

For the frame discretization, solid brick elements C3D8RH were used, with a regular pattern and four layers of elements across the thickness. Hybrid elements were defined to represent nearly incompressible hyper-elastic materials. In addition, hybrid elements overcome the volume strain ‘locking’ problem. This mesh configuration was selected after a comprehensive convergence study, which showed that any further mesh refinement produced minimal variations in the stress distribution.

Triangular shapes are more challenging to test experimentally, as they require defining a custom test set-up. On the other hand, for rectangular shapes a standard test set-up can be employed and interpretation of results is more intuitive. For this reason, a rectangular shape model was chosen for validation purposes.

A typical analysis, carried out for each FE model, consisted of two steps with different types of loading: Step 1, in which a dead load was applied, and Step 2 with the load imposed by varying temperature or external pressure. The purpose of Step 2 was to consider atmospheric loads, i.e., changes in temperature and ambient pressure. When the outside pressure is higher than the pressure of the fill gas inside the unit the panes will recess inward forming a concave shape. In an opposite situation, the glass panes will bulge outward forming a convex shape. The impact of the pressure gradient on the strain increase rate was investigated using the relevant meteorological data for the area of Warsaw, Poland.

Different boundary conditions and loading scheme were applied only in the validation test (Figure 16). An experiment was conducted with an identical glass unit removed from the building. The unit was simply supported at two ends only. First, load cycles (0–400 N) were applied at the midspan. Next, the unit was unloaded, followed by monotonic loading continued up to failure.

Two main parameters were evaluated in the simulations: stress concentration in the seal and strain distribution in both panes. It is worth noting, at this point, that the main goal of these simulations was to obtain a better insight into IGU behaviour and to validate or reject some hypotheses. Therefore, the properties of the different materials and their potential inaccuracy do not compromise the validity of the subsequently derived conclusions.

## 4. Results

In order to evaluate the relative accuracy of representation of the main physical behaviour by the proposed finite element model, it was subjected to the model validation process. In this process the response predictions were compared with the experimental data. The load-displacement relationship obtained from the experimental tests was compared with that obtained from the calculations (Figure 17). The test procedure consisted of two steps: Step 1, in which F = 400 N force was applied in cycles including unloading phase, and Step 2 with monotonic loading continued up to failure. The numerical model gives a fairly accurate representation of Step 1 when the displacements are moderate (marked with a green rectangle on the graph below). Discrepancies are observed in Step 2 when a greater load was applied, probably due to idealizations of the model. The greatest difference was observed at the glass pane breaking load of F = 800 N. The above observations allow us to conclude that the chosen modelling methodology is adequate in terms of the behaviour of the elements subjected to the normal service conditions. On the other hand, it fails to accurately represent the failure (breaking) phase. However, such states are beyond the scope of this study.

Next the units were analysed numerically, taking into account the combined effect of the applied dead load and ambient conditions. More attention was paid to the deformations of elastomer elements, as the primary objective was to determine the effect of loading on the behaviour of the spacer frame (silicone + seal).

The influence of additional environmental loads was assessed using the FE approach presented in Section 3.3 above. Given a set of IGU geometries and the reference production temperature TP = 20 °C, a temperature gradient T > 0 was applied to the gas infill of the cavity. This led to increase of the cavity volume causing additional deformation of the glass panes.

Further FE incremental results are reported to assess the influence of negative/positive pressure gradients on the deformation magnitude. In the case of negative gradient the glass pane and the seal are subjected to suction. The focus was placed on identifying points of the greatest deformations to determine where the strain of glass may affect the seal. Contour maps of maximum principal strains in the top and bottom glass panels are presented in Figure 18 below. Surprisingly enough, the points of the greatest strains (deformation of the glass pane adjacent to the seal) are not located in the middle of the span, but in the places pointed to by arrows.

A similar analysis was performed for the positive pressure gradient, under which the panes bulge outward, because the ambient pressure is lower than the pressure of argon filling the cavity of the unit. The strain values were smaller (in correspondence to a lower absolute value of the applied pressure), but the observations regarding the characteristic points on the IGU perimeter were similar. In both cases tensile strains of the butyl-rubber seal were concentrated on its interface, which when combined with a locally not sufficient bond between PIB and silicone can cause breaking of the seal and sucking it into the unit.

Figure 19 below gives a detailed insight into the strain characteristics of the glazing unit seal. Two frame sections were chosen for comparison, which were virtually ‘cut out’ from the middle of triangle frame’s hypotenuse (a) and shorter side (b), as shown in Figure 19 below. The values were different yet are consistent with the observations made before (compare Figure 18). In both cases the maximum principal strains develop at the connection between silicone and butyl rubber. It should be noted that the glass/silicone/butyl rubber interaction is simplified by assuming ideal bonding conditions. This approach was necessary due to the size of the model and the assumptions made in the calculations.

## 5. Discussion

This article evaluates the mechanical performance of insulating glass units, a system used, for example, on the building facades. The evaluation included both in-situ tests and numerical analyses using Finite Element (FE) models. Refined FE models were developed for the purposes of this research, to obtain a true representation of the actual behaviour of IGUs subjected to various environmental loads, taking into account the influence of the air cavity. This allowed for considering all the relevant phenomena occurring in IGUs during service. The modelling methodology was verified and successfully validated by laboratory testing. The research and FE simulations discussed in this paper allow us to draw the following conclusions:The calculations carried out for triangular units gave the characteristic points of the greatest tensile strains on the unit perimeter. These points are the most probable locations of separation between the PIB seal and silicone, after which the PIB seal becomes displaced inward the cavity. Interestingly enough, these points are not located at the middle of the span, as one would expect, but at intermediate points, as shown in the drawings. These locations are supported by the results of the in situ inspection.The greatest principal strains occurred at the silicone/butyl rubber interface. This should be considered the place that is most susceptible to failure. When insufficient bond strength comes into play, the two materials will come apart at these points. The significance of this phenomenon was reported also in other studies, for example in [11].A constant pressure of argon was assumed. The effect of external factors was checked in two cases: with the outside pressure higher (negative gradient) and lower (positive gradient) than the pressure of the fill gas in the cavity. Negative gradient results in suction applied to the glass panes and to the seal. Positive gradient causes outward bulging of the glass panes. Both situations cause development of tensile strains in the silicone and butyl rubber, which can lead to debonding and displacement inward the cavity.With insufficient adhesion between the silicone and PIB sealants, repeated strain cycles due to changes in ambient pressure or temperature may lead to seal displacement. Therefore, frequent in-process testing should become a priority during IGU production. This is particularly important for units that will be exposed to high temperatures due to sunlight, i.e., facing south-west. The peel test can be carried out for this purpose, for example according to ASTM C794-10 [28] as recommended in [11].Over 95% of glazing units remained tight despite visible damage to the materials of the warm edge system layers. Thus, they still meet the conditions of safe use. As long as the glazing unit remains tight, the surface stresses induced in the outer glass pane due to the action of wind, snow and other factors are transferred to the inner pane by the fill gas. However, when the warm edge system becomes unsealed, the applied load must be carried solely by the outer pane. As a result, the unit loses its initial load capacity and must be replaced immediately.After ten years in service, the percentage of damaged units is considerable and thus warm edge glazing systems must be inspected for damage. The inspection should cover both the effect of damage on the appearance and loss of integrity of sealing and unsealed units should be replaced. The warm edge structure can be initially screened with thermal imaging technique to detect unsealed panels based on compromised thermal performance and inspected visually for the presence of condensate (or traces of condensation) on the inner surfaces of the glass panes.From the standpoint of both durability and appearance (i.e., no displacement of the PIB seal), dual silicone/PIB should be phased out in favour of by stiffer seals including aluminium or recently launched Super Spacer dual seal warm edge spacers [14]. This type of sealing ensures a durable bond to glass and adequate thermal performance, which does not decrease over time.

## Figures and Tables

**Figure 1 materials-14-03594-f001:**
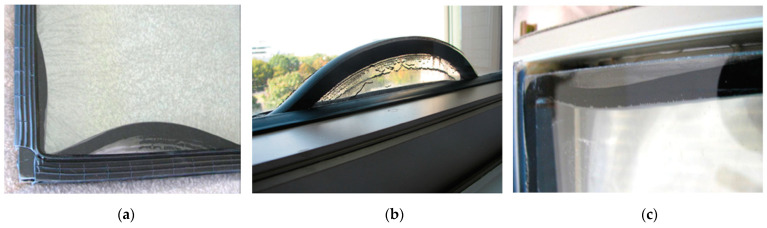
Examples of the IGUs with a PIB dual-seal showing considerable displacement of the IGU spacer seal assembly (reproduced, with permission from [11], copyright ASTM International, 100 Barr Harbor Drive, West Conshohocken, PA 19428).

**Figure 2 materials-14-03594-f002:**
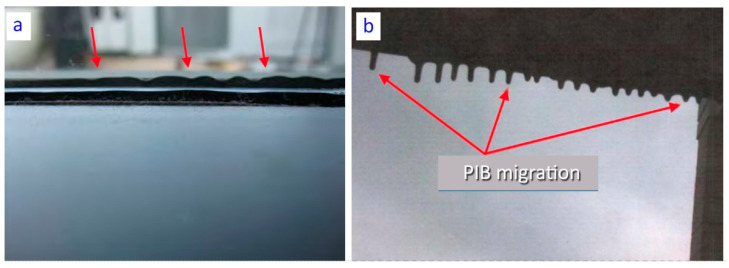
Examples of PIB failures: (**a**) squeeze out or “creep”, (**b**) migration [12].

**Figure 3 materials-14-03594-f003:**
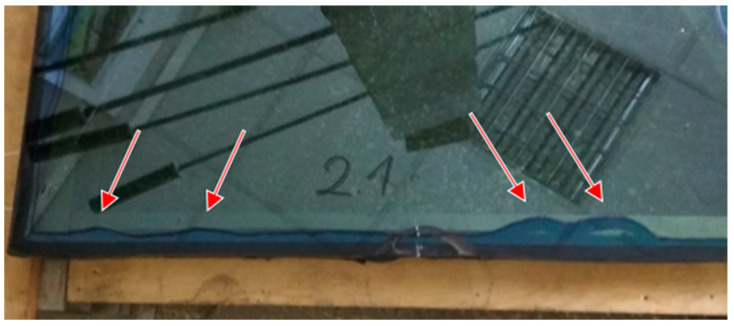
Examples of composite PIB seal displacement affecting the silicone-butyl connection and probably also the seal function.

**Figure 4 materials-14-03594-f004:**
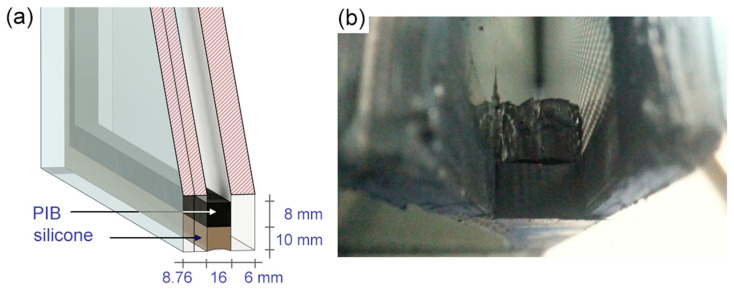
Overview of warm edge system: (**a**) schematic of the model (**b**) photo of the actual cavity space with primary and secondary seals.

**Figure 5 materials-14-03594-f005:**
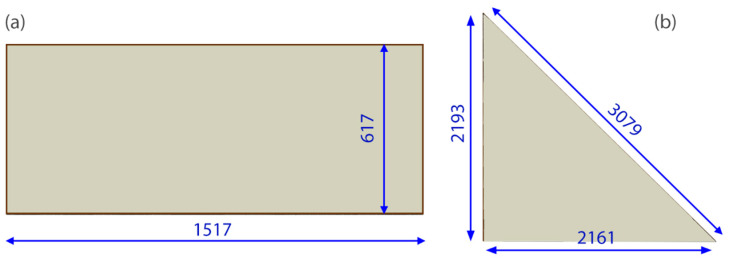
Dimensions of IGU specimens, mm (**a**) rectangular and (**b**) triangular.

**Figure 6 materials-14-03594-f006:**
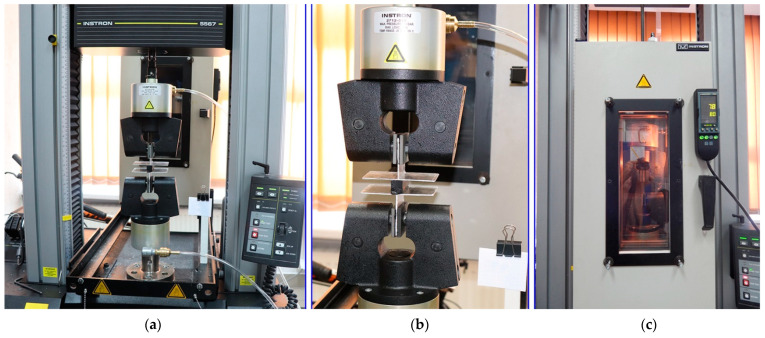
Determination of the properties of materials: (**a**) test frame; (**b**) pneumatic jaws; (**c**) environmental chamber.

**Figure 7 materials-14-03594-f007:**
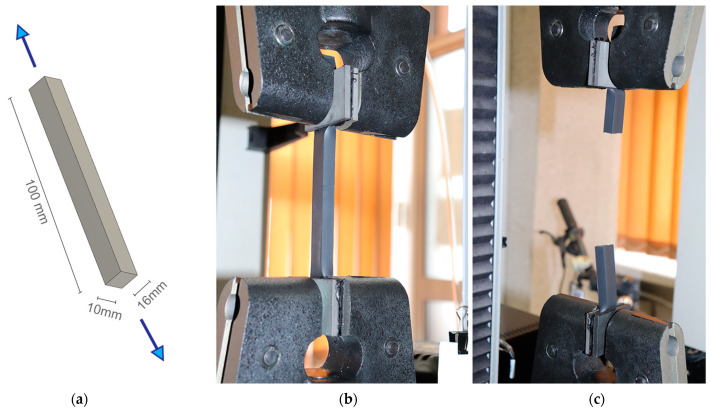
Testing of silicone: (**a**) dimensions of cut out specimens (**b**); tensile test at 23 °C; (**c**) finish after sample break.

**Figure 8 materials-14-03594-f008:**
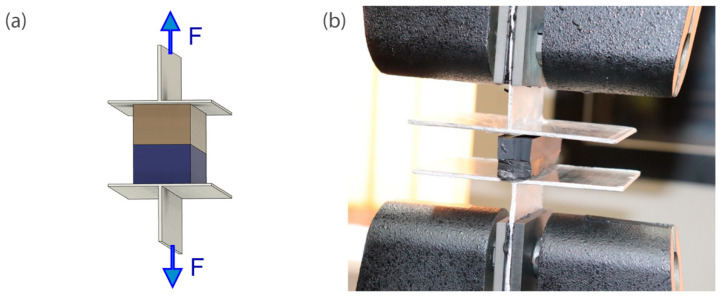
Detachment of PIB and silicone: (**a**) test set-up: (**b**) mounting of the specimen in the strength testing machine.

**Figure 9 materials-14-03594-f009:**
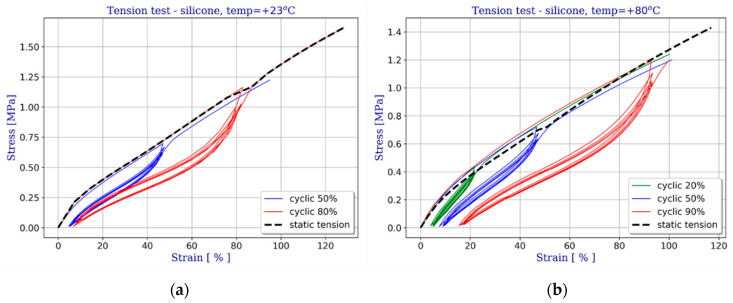
Results of uniaxial cyclic tension tests at different stress levels and different temperatures: (**a**) T = 23 °C; (**b**) T = 80 °C.

**Figure 10 materials-14-03594-f010:**
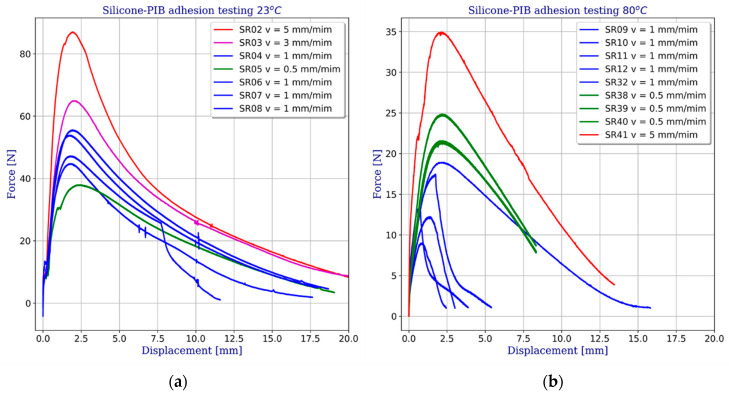
Force-displacement results obtained on silicone-butyl specimens: (**a**) at 23 °C, (**b**) at 80 °C.

**Figure 11 materials-14-03594-f011:**
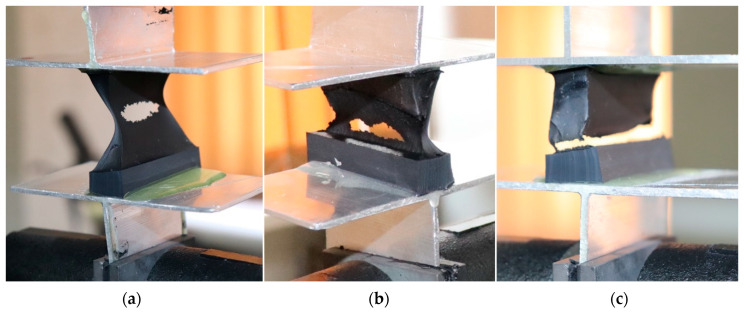
Specimen failure mechanisms at 23 °C: (**a**) PIB viscoelastic flow and large deformations; (**b**) mixed adhesion and cohesive failure; (**c**) adhesive failure PIB-silicone.

**Figure 12 materials-14-03594-f012:**
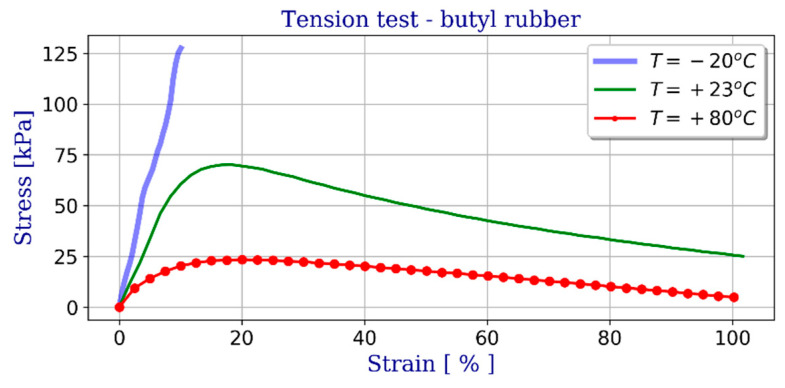
Butyl rubber (PIB) uniaxial tension results (engineering stress–strain relationship).

**Figure 13 materials-14-03594-f013:**
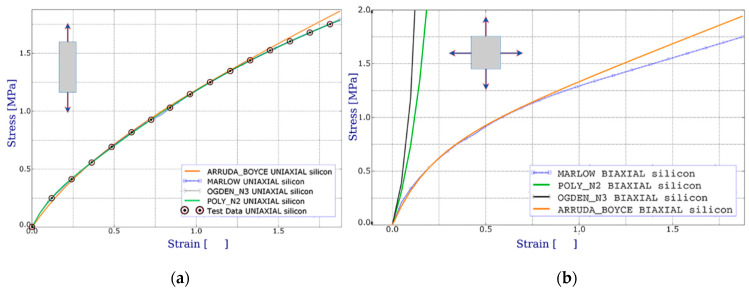
Comparison of the test data and various hyperelastic models. Stress–strain curves for tension test: (**a**) uniaxial; (**b**) biaxial.

**Figure 14 materials-14-03594-f014:**
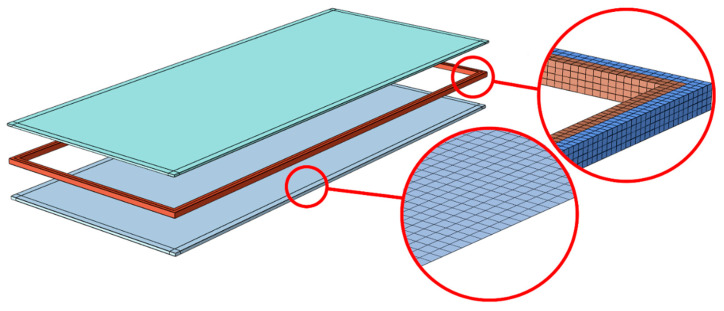
Overview of the rectangular unit under analysis–geometry, make-up and example meshing.

**Figure 15 materials-14-03594-f015:**
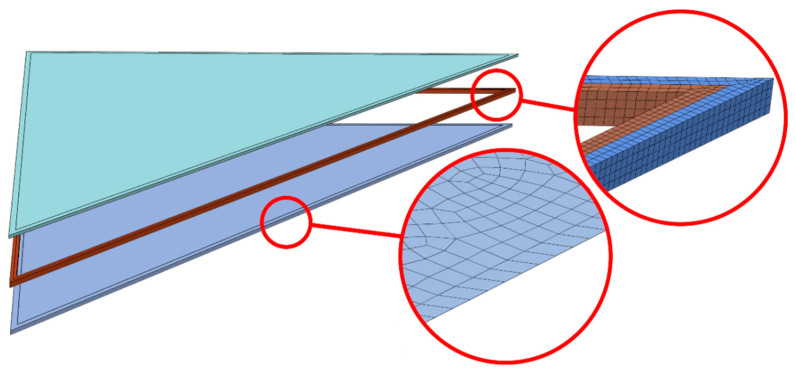
Overview of the triangular unit under analysis–geometry, make-up and example meshing.

**Figure 16 materials-14-03594-f016:**
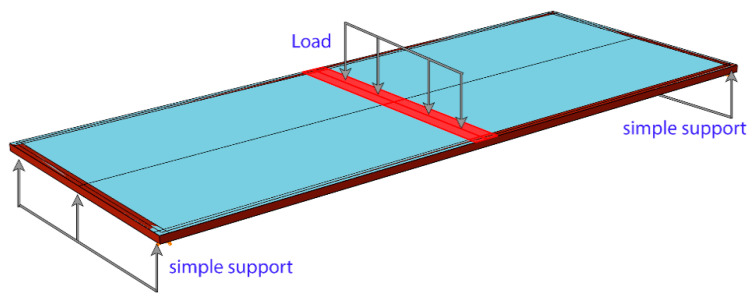
Boundary conditions and loading scheme during the rectangular panel validation test.

**Figure 17 materials-14-03594-f017:**
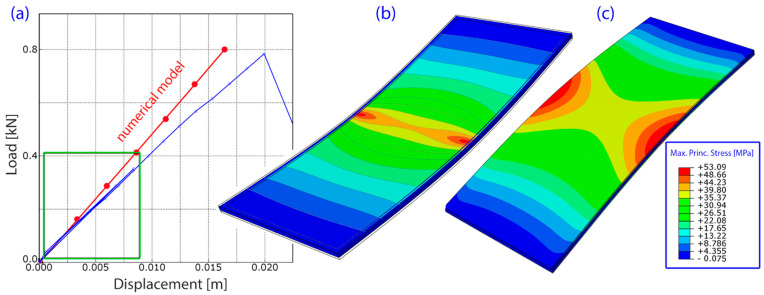
Rectangular model validation test: (**a**) equilibrium path (load-displacement); (**b**) contour plots showing the maximum principal stress in the top panes; (**c**) maximum principal stress in the bottom panes.

**Figure 18 materials-14-03594-f018:**
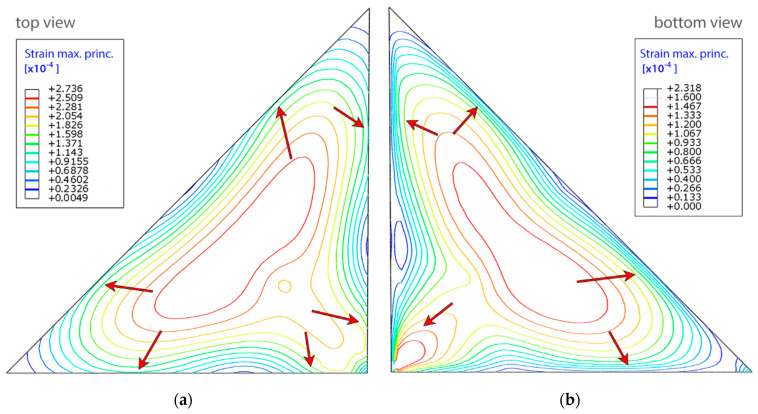
Contour maps of the distribution of the maximum principal strains induced by a negative pressure gradient: (**a**) view from the top; (**b**) view from the bottom on panes separated by a PVB interlayer.

**Figure 19 materials-14-03594-f019:**
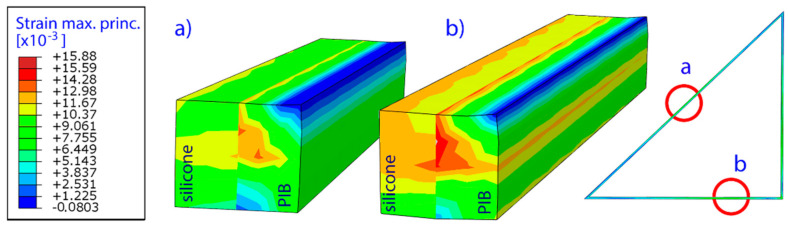
Cross-sectional view showing the distribution of the maximum principal strains in two selected pieces of the frame (silicone and butyl rubber PIB). Locations (**a**) and (**b**) presented on the right subfigure.

**Table 1 materials-14-03594-t001:** Silicone specimens subjected to the tensile test (left) and to tear test (right).

Test of Silicone				Test of Silicone + PIB			
Temp. °C	Rate [mm/min]	Number of Specimens	SD Gauge [mm]	Temp. °C	Rate [mm/min]	Number of Specimens	SD Gauge [mm]
−20	35	5	3.11	−20	1	5	0.62
23	70	6	4.58	23	35	1	-
23	35	2	1.41	23	5	1	-
80	35	5	1.52	23	3	1	-
80	70	10	2.73	23	1	3	0.67
				23	0.5	1	-
				80	5	8	2.15
				80	1	14	0.99
				80	0.5	6	1.64

**Table 2 materials-14-03594-t002:** Properties of fill gases [25].

Name	Unit	Air	Argon	Krypton
Thermal conductivity L [W/(mK)] λ=a+b T(K)	a [W/(mK)]	2.873 × 10^−3^	2.285 × 10^−3^	9.443 × 10^−4^
	b [W/(mK^2^)]	7.760 × 10^−5^	5.149 × 10^−5^	2.826 × 10^−5^
Dynamic viscosity m [Pa s] μ=a+b T(K)	a [Ns/m^2^]	3.723 × 10^−6^	3.379 × 10^−6^	2.213 × 10^−6^
	b [Ns/m^2^K]	4.940 × 10^−8^	6.451 × 10^−8^	7.777 × 10^−8^
Heat capacity *cp* [J/(kgK)] $c_{p}=a+b\;T(K)$	a [J/(kgK)]	1002.737	521.929	248.091
	b [J/(kgK^2^)]	1.232 × 10^−2^	0	0
Mass [kg/mol]		28.970 × 10^−3^	39.948 × 10^−3^	83.80 × 10^−3^

## Data Availability

All data included in this study are available upon request by contact with the corresponding author.

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
