# Peer review of "Polyisobutylene and Silicone in Warm Edge Glazing Systems—Evaluation of Long-Term Performance"

_materials, 2021, doi:10.3390/ma14133594_

Round 1

Reviewer 1 Report

The paper's topic is very interesting but the paper contains several flaws/ misinterpretations that must be corrected, references are missing and thus the paper cannot be accepted in the current form:

Chapter 1 / Introduction is too long for a journal paper and must be shortened and specific aspects must be more specific:

line 82: must be "higher permeability", not "lower"

line 91: resistance of what?

lines 98-101: references missing

lines 106-108: references/ reports/ justification of the statement missing

line 127: "prone to displacement" under which specific loading?

line 139: in which direction acts the pressure during the assembly and is the pressure applied?

line 140-142: Underpressure conditions in the cavity (long time) can results in such damages? It seems to be rather an incompatibility of materials.

line 159: PIB typically exhibits hyperelastic or hyper-viscoplastic material behavior and a Young's modulus as one parameter cannot be defined. Maybe the share "stiffness" is better here.

line 161: "mechanical interaction" must be explained in more detail in the context of incompressibility of the PIB and the adjacent sealant and the lateral contraction.

line 167: "weaker" in which regard?

line 179: Spacers are typically NOT rigid.

line 189: one of too much

line 192: Is "buckling" here the right word?

line 203-204: what is the reference for this statement? total amount of curtain wall units? Is there a difference between 4500 IGUs and curtain IGU? 

line 220: Please show a overview picture (thermografic) of the measured facade. 

Chapter 2

line 261: add references 

line 282-286: The information about the geometrical properties must be also included in Fig. 4.

line 284: the product name of the PIB is missing 

Fig. 4: In this Figure, the laminate should be indicated and it is unclear what is shown in Fig. 4 (b).

Figure 5: Dimension on the triangular picture missing.

Chapter 2.2 could be reduced to the most relevant information.

Please explain: Which properties are to be determined? Based on which standards or methods? Please add a schematic image of the test setup. I don't understand really the test direction and arrangement of the two components (It seems to be a combination of PIB and silicone in figure 6b) . Please describe the reasons for such a test setup.

Figure 6: The test specimen and its geometrical properties and the scattering of the dimensions is missing and must be added in a Figure before showing it in the test assembly.

Figure 7: It is unclear if really all samples failed in the centre here as the local fixing strongly influences the material behavior and stress/ strain  distribution, also the samples geometry lets expect different origins of failure. Tension tests are usually carry out by a dogbone geometry. Here you used a bar geometry, please explain why. 

Figure 8: Wrong caption. Also, the explanation of the geometry of the specimen is missing.

line 354/ Figure 9: Where are the results of -20°C? 

Figure 9: An explanation is missing what happens when the blue line is re-loaded above the initial stress-strain level and why it was done here and not with the red line.

line 375-376: No. A difference can be seen from Fig.9

line 380: What did you do with the specimen that broke at the fixing as these cannot be used to analyze a failure criterion?

line 395/ 398: Figure 10 does not show plasticity but damage.

line 413: Where are the results given? I don't see any strength values? only a graph force - displacement in Figure 10.

Figure 11 (a), (b) does not show plasticity but viscoelasticity and large deformations and cohesive failure (silicone), mixed failure adhesion and cohesive failure (Silicone/ PIB) or adhesive failure PIB-silicone? (c) shows that the original geometry is almost constant so it is damage, not plasticity.

line 459: It is unclear how the parameters for the PVB where determined.

line 461-464 is a repetition

Figure 12 seems to show engineering stress/ strain and the true stress/ strain should be shown as well.

line 474-475: which tests were done and how could you distinguish between plasticity and damage which seems to govern here?

line 476 what is HMH? Reference?

line 486-495 should be shortened

line 501: I do not agree, see for example: Rosendahl et al. (2019) Materials & Design, Vol. 182, 108057

equation (2): det, not def

line 518: .... invariant of the left Cathy-Green strain tensor. 

equation (4) is wrong, it is the sum of the squared stretches

Figure 13: parameters of the different models are missing

line 526-533: Which of the models discussed before is used now?

Figure 17: How do you explain the difference between simulation and experiment?

line 636: This load case is not as relevant for IGU as the pressure load case with effects on sealant in the edge, so why was it chosen here?

line 641:  which test do you refer to here? refer to the experimental test (section ?) 

line 667: glass strains are very small and irrelevant for the sealant

line 677: how do you define "large" and "poor"?

Figure 19: The mesh is not shown and must be added, it seems that the meshing was not small enough. 

line 696: IGU is not a material

line 701-702: The comparison between FE and section 2 is missing.

line 710: references/ examples for the definition of the locations for this statement are missing in the report. 

line 743-745: This statement is not corroborated by the results shown in the paper.

Generally, the conclusions and interpretations given here and the rest of the paper do not really fit and should be changed according to the results shown.

Reviewer 2 Report

The strength of this paper lies in the fieldwork study carried out on the IGU system. The authors produce an evidence-base for the type of failure in IGU units whilst their service.  

The mechanical characterisation of sealing materials is standard. They have been widely utilised to develop a numerical model to predict stress concentration in the seal and strain distribution in both panes. However, it would be beneficial for the reader to relate the potential cause of the real-time damage in IGU systems by using their finite element study.

Also, authors need to clarify whether they are used in pristine or extracted materials from their field study to fabricate the IGU system tested in this study. No information is given about the supplier or grades of the materials used to develop the IGU unit.

Table 1 shows several samples were tested for the same conditions. What is the standard deviation for these tests?

The abstract of this paper is very generic. This should include the main outcome of the study.

There are few typos, see below for example

  • Line number 324-325: Correct the usage of the degree Celsius symbol
  • Line number 328: Table 1 Correct the usage of the degree Celsius symbol
  • Line number 395-396: Correct the usage of the degree Celsius symbol
  • Line number 401-402: Correct the usage of the degree Celsius symbol
